# Electron-Induced Decomposition of Uracil-5-yl *O-*(*N*,*N*-dimethylsulfamate): Role of Methylation in Molecular Stability

**DOI:** 10.3390/ijms22052344

**Published:** 2021-02-26

**Authors:** Eugene Arthur-Baidoo, Karina Falkiewicz, Lidia Chomicz-Mańka, Anna Czaja, Sebastian Demkowicz, Karol Biernacki, Witold Kozak, Janusz Rak, Stephan Denifl

**Affiliations:** 1Institute for Ion Physics and Applied Physics, University of Innsbruck, Technikerstrasse 25/3, 6020 Innsbruck, Austria; Eugene.Arthur-Baidoo@uibk.ac.at; 2Center for Biomolecular Sciences Innsbruck, University of Innsbruck, Technikerstrasse 25/3, 6020 Innsbruck, Austria; 3Department of Physical Chemistry, Faculty of Chemistry, University of Gdańsk, Wita Stwosza 63, 80-308 Gdańsk, Poland; karina.falkiewicz@phdstud.ug.edu.pl (K.F.); lidia.chomicz-manka@ug.edu.pl (L.C.-M.); anna.czaja@phdstud.ug.edu.pl (A.C.); davelombardo@wp.pl (W.K.); 4Department of Organic Chemistry, Faculty of Chemistry, Gdańsk University of Technology, Narutowicza 11/12, 80-233 Gdańsk, Poland; sebastian.demkowicz@pg.edu.pl (S.D.); karol.biernacki@pg.edu.pl (K.B.)

**Keywords:** uracil derivatives, gas phase, radiosensitizers, low-energy electrons, electron attachment, resonance energy

## Abstract

The incorporation of modified uracil derivatives into DNA leads to the formation of radical species that induce DNA damage. Molecules of this class have been suggested as radiosensitizers and are still under investigation. In this study, we present the results of dissociative electron attachment to uracil-5-yl *O*-(*N*,*N*-dimethylsulfamate) in the gas phase. We observed the formation of 10 fragment anions in the studied range of electron energies from 0–12 eV. Most of the anions were predominantly formed at the electron energy of about 0 eV. The fragmentation paths were analogous to those observed in uracil-5-yl *O*-sulfamate, i.e., the methylation did not affect certain bond cleavages (O-C, S-O and S-N), although relative intensities differed. The experimental results are supported by quantum chemical calculations performed at the M06-2X/aug-cc-pVTZ level of theory. Furthermore, a resonance stabilization method was used to theoretically predict the resonance positions of the fragment anions O^−^ and CH_3_^−^.

## 1. Introduction

Radiotherapy is used in the treatment of more than 50% of all cancer patients [1]. However, the specificity of the tumor environment such as a low concentration of oxygen (hypoxia) impedes the efficacy of this modality. Hypoxia—a hallmark of solid tumors—causes tumor cells to become resistant to ionizing radiation (IR) [2]. Therefore, an obvious way of improving radiotherapy is to introduce chemical agents, known as radiosensitizers, that make cancer cells more sensitive to IR [3,4]. Nitroimidazoles such as nimorazole, are one class of such substances that have demonstrated their radiosensitization potential as oxygen mimetics [5]. Other derivatives such as fluorinated pyrimidines and heterocyclic-*N*-oxides are being exploited to counter the setbacks of hypoxia and improve the treatment [6,7,8,9].

Radiation therapy employs IR to deposit energy in the biological medium. As the radiation traverses the medium, a large percentage of the deposited energy is channeled into the production of low-energy electrons with kinetic energies below 30 eV and the maximum of energy distribution around 9–10 eV [10]. These so-called ballistic electrons are known to induce DNA damage like single- and double-strand breaks, base release and sugar modifications [11,12,13] when DNA is bombarded under an ultra-high vacuum. One of the underlying ways of inducing such damage, i.e., dissociative electron attachment (DEA), is a selective mechanism in terms of bond cleavage in a molecule. In the DEA process, the attachment of a low-energy electron results in the formation of a temporary negative ionic (TNI) state, which may dissociate into an anion and neutral radical [14] as shown in Reaction (1):e^−^ + AB → (AB)*^−^ → A^−^ + B^⦁^(1)
where AB is the parent molecule, (AB)*^−^ represents the TNI and the A¯/B^⦁^ stands for the fragment anion and the corresponding neutral radical, respectively.

It has been shown that the presence of high electron-affinity substituents in nucleobases increases their susceptibility towards DEA. Halogenated uracil derivatives such as 5-bromouracil (BrU), 5-chlorouracil, 5-fluorouracil and 5-iodouracil, which have been extensively studied, demonstrated such behavior [15,16,17,18,19]. It was also shown that the substitution of hydrogen with a bromine atom at the C5 position in nucleobases increases the electrophilic properties of compounds [20]. Rak and co-workers have suggested several 5-substituted pyrimidine derivatives as potential radiosensitizers of hypoxic cells. These molecules comprise 5-trifluoromethanesulfonyl-uracil (OTfU), uracil-5-yl *O*-sulfamate (SU) and 5-selenocyanatouracil (SeCNU), and have been studied both theoretically and experimentally through stationary radiolysis in an aqueous solution and crossed electron-molecular beam (CEMB) experiments in the gas phase [21,22,23]. While SeCNU and OTfU could work as potential radiosensitizers, SU was dubbed an “illusive radiosensitizer” which, contrary to expectations, is not prone to the DEA process in water [22].

It has been suggested that, to enhance the DNA response to radiation, a modified nucleoside incorporated into DNA must be susceptible to DEA. The latter process leads to the production of a reactive radical of the nucleobase inside the DNA strand [24]. In uracil derivatives, like BrU, the loss of the substituent leads to the formation of the uracil-5-yl (U-H)^•^ radical, which, by the abstraction of a hydrogen atom from a nearby deoxyribose residue, starts the subsequent radical reactions that ultimately produce DNA strand breaks [25,26] (it was demonstrated that for the abstraction of any hydrogen atom from the sugar moiety in DNA at least one pathway exists that forms a DNA strand break [26]). Thus, it seems important to study the DEA process for the compounds that are expected to have radiosensitizing properties. A gas phase study on the interactions of the proposed molecules with low-energy electrons enables the necessary information about the fragmentation pathways and the mechanism of anion formation in the DEA process to be obtained. Such information deepens our understanding of the mechanism of radiosensitization triggered by low-energy electrons.

The recent DEA studies with SU (Figure 1a) demonstrated low-energy electron-induced decomposition yielding various anionic and radical products due to simple and complex fragmentation pathways upon electron attachment to SU in the gas phase [22]. The results of the CEMB experiments were supported by quantum chemical calculations, which justified the assumed pathways of the DEA processes. However, steady-state radiolysis in water shows that these DEA channels are quenched irrespectively of pH or incident radiation dose, demonstrating unexpected stability of SU against electron attachment in an aqueous solution [22]. These findings highlighted the limitations of the methodology used to determine whether the molecule under investigation can be classified as a radiosensitizer.

Herein, we report the results of CEMB experiments on the anion formation and fragmentation pathways triggered by 0–12 eV electron attachment to uracil-5-yl *O*-(*N*,*N*-dimethylsulfamate) (DMSU; Figure 1b). DMSU is a dimethylated form of SU in which the hydrogen atoms of the amino group in the substituent were replaced by two methyl groups. We synthesized the dimethyl derivative of SU, since introducing methyl groups into the molecular structure of SU allows us to examine whether methylation influences the dynamics of DEA. Moreover, the O^—^ and NH_2_^—^ fragment anions might form in the CEMB experiments for SU carried out recently [22]. The mass resolution of the mass spectrometer utilized did not allow differentiation between these isobaric species. The present results of analogous measurements for DMSU should dispel any doubts concerning the fragmentation of SU. The experimental results are supported by quantum chemical calculations on the thermodynamic thresholds of fragmentation reactions and resonance positions.

## 2. Results

The interaction of low-energy electrons with DMSU (C_6_H_9_N_3_O_5_S) resulted in the experimental detection of ten fragment anions with most of them formed within the energy range below 2 eV. The peak positions and experimental onsets as well as the thermodynamic threshold for the various reaction pathways are presented in Table 1. The anion yields as a function of the electron energy for each fragment anion are shown in Figure 2, Figure 3 and Figure 4. As in our previous studies on the modified pyrimidine derivatives, the presence of the parent anion and the dehydrogenated parent anion was not observed [21,22]. Regardless of the side group attached to the UOSO moiety, these compounds exhibit characteristic fragmentation pathways due to the C-O, S-O and S-X (where X = NH_2_ [22], CF_3_ [21]) bond cleavages leading to the formation of several negative ions.

In our electron beam experiment, we observed a fragment anion with mass 191 u, which we assigned to the structural formula UOSO_2_^−^. The pathway to the anion may follow a simple S-N bond cleavage in the parent compound and may proceed via Reaction (2) resulting in the release of the *N*,*N*-dimethyl group CH_3_NCH_3_^⦁^ as its corresponding neutral.
e^−^ + DMSU → (DMSU)*^−^ → UOSO_2_^−^ + CH_3_NCH_3_^⦁^(2)

Figure 2a shows the anion efficiency curve for UOSO_2_^−^ with a sharp resonance at ~0 eV and a small shoulder around 0.1 eV. Quantum chemical calculations predict that the anion formation proceeds via an exothermic reaction with a thermodynamic threshold of −1.78 eV at 386 K, which agrees well with the experimental onset of 0 eV. A relatively large AEA of 4.11 eV for electron attachment to the UOSO_2_ was also calculated (see Table 1).

We also obtained fragment anions with masses 127 and 126 u, which may form upon the Reactions (3) and (4):e^−^ + DMSU → (DMSU)*^−^ → UO^−^ + CH_3_NCH_3_SO_2_^⦁^(3)
e^−^ + DMSU → (DMSU)*^−^→ (UO-H)^−^ + CH_3_NCH_3_SO_2_H(4)

Figure 3b,c presents the anion efficiency curve for the formation of anions with masses 127 and 126 u, respectively. For 127 u, we obtained a sharp peak at ~0 eV, another resonance at 0.3 eV and a broad resonance around 1.3 eV. Similar resonance positions were observed upon DEA to SU [22], indicating that the presence of methyl groups does not interfere with the anion formation processes. The adiabatic electron affinity (AEA) of UO was determined to be 2.40 eV which is close to 2.38 eV predicted in the case of OTfU [21]. A further dehydrogenation of UO^−^ resulted in the formation of (UO-H)^−^ with mass 126 u. The (UO-H)^−^ anion was recorded as the most abundant fragment anion formed upon DEA to DMSU. To compute the thermodynamic threshold, we considered two sites for the loss of H, i.e., N3-H (−0.78 eV, Table 1) and N1-H (−1.41 eV, Table 1). It was shown in a previous DEA study with uracil that the N1-H site is a more thermodynamically favored channel to contribute to dehydrogenation [28]. Therefore, we propose that the additional H loss occurs at the N1-H position. The AEA of (UO-H) was predicted to be 4.84 eV. Both UO^−^ and (UO-H)^−^ anions are detected with an experimental onset close to 0 eV. According to our thermodynamic calculations, exothermic reactions (Reactions (3) and (4)) with a threshold of −0.95 eV were predicted for UO^−^ and −1.41 eV for (UO-H)^−^, respectively, which agree with the experimental results. The cleavage of the S-O bond also leads to the complementary anion CH_3_NCH_3_SO_2_^−^ with mass 108 u. Figure 2d shows the anion efficiency curve for CH_3_NCH_3_SO_2_^−^ which is formed via Reaction (5):e^−^ + DMSU → (DMSU)*^−^ → UO^⦁^ + CH_3_NCH_3_SO_2_^−^(5)

Similar behavior involving S-O bond cleavage was observed for SU and OTfU, which led to the release of the corresponding anions NH_2_SO_2_^−^ and CF_3_SO_2_^−^ in DEA to these respective molecules [21,22]. The presence of similar resonances below 2 eV suggests that the mentioned anions associated with DEA to SU and OTfU may be formed from the same TNI state of the parent molecule.

In addition, we observed the anion with mass 124 u, which was assigned to CH_3_NCH_3_SO_3_^−^ accompanied by the release of a uracil-5-yl radical as its counterpart neutral fragment. The formation of the fragment anion proceeds via the exothermic Reaction (6) with a predicted thermodynamic threshold of −0.64 eV.
e^−^ + DMSU → (DMSU)*^−^→ (U-H)^⦁^ + CH_3_NCH_3_OSO_2_^−^(6)
e^−^ + DMSU → (DMSU)*^−^→ U + CH_3_NCH_2_OSO_2_^−^(7)

Three resonance positions were observed for 124 u with an experimental onset of 0 eV, as shown in Figure 3a. The anion yield curve for CH_3_NCH_3_OSO_2_^−^ also exhibits a resonance at 0.3 eV and a broad resonance structure at around 1.4 eV. The theoretical AEA for CH_3_NCH_3_OSO_2_ was predicted to be 4.66 eV. Similarly, to the anions originating from the S-O bond cleavage, we observed also the loss of a single hydrogen atom via Reaction (7) with an experimental threshold of 0 eV. Computationally, we determined the threshold for the formation of CH_3_NCH_2_OSO_2_^−^ to be −1.79 eV and an adiabatic electron affinity to be 4.88 eV.

Electron attachment to a molecule aside from single bond cleavages can result in multiple bond cleavages and complex rearrangement in the molecule. We detected anions due to both the N-S and S-O bond cleavages that may occur via either within the parent molecule or from subsequent dissociation of CH_3_NCH_3_SO_3_^−^ leading to the formation of two anions with masses 80 and 64 u. The two anions appear as the second and third most abundant ones upon DEA to DMSU.
e^−^ + DMSU → (DMSU)*^−^ → SO_3_^−^ + CH_3_NCH_3_U(8)
e^−^ + DMSU → (DMSU)*^−^ → SO_2_^−^ + CH_3_NCH_3_OU(9)

From Reaction (8) and based on the molecular structure of DMSU, the anion with mass 80 u corresponds to SO_3_^−^. According to Figure 3d, the anion yield curve exhibits a sharp peak at 0 eV with a shoulder around 0.3 eV and a further broad resonance at 1.3 eV. The calculated thermodynamic threshold was found to be −2.09 eV, which agrees with the experimental threshold of 0 eV. The AEA of SO_3_ was found to be 5.09 eV. At mass 64 u, we detected another anion structurally assigned to SO_2_^−^ following the loss of an oxygen atom from SO_3_^−^. With an experimental onset near 0 eV, the thermodynamic threshold of the exothermic Reaction (9) leading to the formation of SO_2_^−^ was found to be −0.66 eV, which agrees well with the experimental value. In our current study, an AEA of 1.43 eV for SO_2_ was derived, which is also comparable with both values from reported literature. Previously, Ameixa et al. [21] reported an AEA value of 1.42 eV for SO_2_, which was below the calculated AEA value using the B3LYP method (1.58 eV) [29]. In DEA to SU, the SO_2_^−^ anion showed two peaks at 1.1 and 5.4 eV [22]. Upon methylation, the intensity of the high-energy resonance near 5.4 eV seems to be suppressed, and the anion yield curve is dominated by a peak at 0 eV. Additionally, the two anions at 80 and 64 u show peaks at similar energies which can be associated with the same TNI states.

Due to the sterical proximity of the methyl groups and the SO_3_ moiety, we investigated the formation of the H_2_SO_3_^−^ and H_2_SO_2_^−^ fragment anions in our quantum chemical calculations. These anions would form by hydrogen transfer from the methyl groups to SO_3_ and SO_2_ groups. It is worth noticing that electron affinities for the H_2_SO_2_ and H_2_SO_3_ fragments are negative at the M06-2X/aug-cc-pVTZ level (see Table 1), suggesting that the respective anions are short-lived states. It is well known that autodetachment of electrons from a short-lived state proceeds in 10^−14^ s in the gas phase [30], which would prevent registration of the respective signal in the experimental setup used in the current work. Therefore, we recalculated the AEA of these fragments using a more reliable G2MP2 method of chemical accuracy. It turned out that both anions, H_2_SO_2_^—^ and H_2_SO_3_^—^, are adiabatically stable at the G2MP2 level (see Table 1). However, we were not able to detect these in the present experiment. Figure 3c,e shows the ion yield recorded at masses 82 and 66 u, respectively. From [31], the natural isotope abundances of SO_3_ and SO_2_ are 100% (80 u), 0.8% (81 u), 5.0% (82 u) and 100% (64 u), 0.8% (65 u), 4.8% (66 u), respectively. From Figure 3c–f, the ratio of the experimentally recorded yields of the masses 80:82 and 64:66 were both determined to be 4.4%, which is in reasonable agreement with the natural isotope composition. Therefore, the measured ion yield for masses 82 and 66 u can be assigned to the isotopes ^34^S^16^O_3_^−^ and ^34^S^16^O_2_^−^, respectively.

For all the above discussed molecular fragmentations upon DEA to DMSU, the major ion yield was observed near 0 eV, which was in agreement with the exothermicity of the corresponding reactions (see negative values of thermodynamic thresholds in Table 1). However, in two cases, the maximum efficiencies were found at higher energies—for mass 16 u at 4.7 and 8.8 eV (interpreted as a release of the oxygen anion, O^−^) and for mass 15 u at 8.5 eV (interpreted as a release of CH_3_^−^ or NH^−^ anion). Figure 4a,b shows the anion efficiency curve for the anion with masses 16 and 15 u. The formation of the O^−^ anion (Figure 4a) is possible via four pathways, related to the cleavages of the following bonds—C2=O7, C4=O8, S10=O11 and S10=O12 (for atom numbering see Figure 1b). Formation of the O^−^ anion from the O9 oxygen atom is less possible as it would demand breaking simultaneously both the C5-O9 and O9-S10 bonds. The endothermic reaction leading to the O^−^ anion proceeds via Reaction (10):e^−^ + DMSU → (DMSU)*^−^ → (CH_3_NCH_3_SO_3_U-O)^⦁^ + O^−^(10)

In order to assign the resonance to the dissociation site, we calculated the thermodynamic thresholds for all four sites, as presented in Table 1. From the predicted values, we conclude that the O^−^ formation at O11 and O12 positions is energetically more favorable than at O7 and O8 positions. As mentioned above, experimentally we obtained two peaks, one located at 4.7 eV and another broad (more abundant) resonance with a maximum at 8.8 eV. In the previous study with SU [22], we observed a similar ion yield at m/z 16; just the ion yield between 4.5 and 6 eV was plateau-like without a minimum as observed in the present data. Such merged contribution in the previous data for SU may be explained by an additional contribution from the isobaric NH_2_^−^, which does not form from the present compounds due to the methylation of the N13.

In the case of the main peak in the O^−^ ion yield at 8.8 eV, all considered reaction pathways are energetically opened. Therefore, an assignment by the thermodynamic threshold was not possible. Thus, we estimated the resonance positions for the different oxygen sites using the charge stabilization method (see computations). The extrapolation of binding energy, D, indicates that a TNI state corresponding to the release of the pyrimidine ring oxygen (O7, O8—see Figure 1b) is formed around 8.3 eV while that leading to the release of oxygen from the sulfamate moiety (O11, O12—see Figure 1b) occurs around 5.0 eV (see Figure 5A,B). These estimates of resonance positions, i.e., 5.0 and 8.3 eV correspond reasonably to the experimental maxima of 4.7 and 8.8 eV, respectively (see Figure 4a).

We note that the presence of a characteristic resonance for O^−^ around 8–9 eV has also been reported in many DEA studies [32,33,34]. In comparison with the literature, Huber et al. [35], in their study on hydroxyurea, calculated a similar thermodynamic threshold of 4.57 eV at room temperature (298 K) for the O^−^ formation using the G4(MP2) method. The adiabatic electron affinity of 1.41 eV for the O atom (see Table 1) is in good agreement with that determined by Rienstra-Kiracofe et al., using photoelectron spectroscopy [29].

The anion formed with mass 15 u can be assigned to either CH_3_^−^ or NH^−^ since our experimental mass resolution does not allow separation of the two isobaric anions. In a study on the amino acid β-alanine, Vizcaino et al. used a double-focusing sector field mass spectrometer experiment and reported two resonance positions around 6 and 9 eV, which were associated with the formation of the anion with mass of 15 u but a clear assignment to either NH^−^ or CH_3_^−^ was not possible. In contrast, the separation of the ion yields for NH^−^ and CH_3_^−^ was achieved in a previous DEA study with glycine [36]. In that study, CH_3_^−^ was predominantly observed at about 6 eV, while NH^−^ showed a main peak at about 11 eV. Since the experimental data for DMSU were not conclusive, we tried to assign the observed signal using charge stabilization calculations. Predicting the resonance responsible for the release of the NH^−^ anion is technically more complicated than estimating the CH_3_^—^ resonance since in the former DEA process two covalent bonds have to be cleaved. Therefore, we calculated the resonance positions corresponding to the release of CH_3_^—^ and obtained 6.21 and 6.42 eV (see Figure 5C) depending on the methyl group. As indicated by Figure 4b and Table 1, the maximum yield of the formation of the anion with mass 15 u is observed for 8.5 eV, thus, 6.3 eV on the average, calculated for the CH_3_^—^ release is much too low to explain the experimental observation. In summary, one can conclude the signal with mass 15 u is associated with freeing the NH^—^ rather than CH_3_^—^ anion.

## 3. Methods

### 3.1. CEMB Experiments

The experimental results presented in this work were obtained using a CEMB experiment discussed previously in detail [37]. Here, we present only a brief description of this experiment. The setup consists of a hemispherical electron monochromator (HEM) coupled with a quadrupole mass analyzer under high vacuum conditions (10^−8^ mbar). The sample was placed in an oven and resistively heated to a temperature of 386 K for sublimation. The molecules in the gas phase were then introduced in the form of an effusive beam into the interaction region via a capillary. In the interaction region, the electron beam produced by a heated filament and directed through the electron monochromator was made to cross perpendicularly with the molecular beam. To ensure the best compromise between the ion beam intensity and energy resolution, 110 meV (at full width at half maximum) was chosen for the electron energy resolution at an electron current of 20–70 nA. The anions formed were extracted towards the entrance of the quadrupole mass spectrometer via a weak electrostatic field. The electron energy scale was calibrated using the well-known 0 eV peak in the ion yield of Cl^−^ from CCl_4_.

### 3.2. Synthesis

Uracil-5-yl *O*-(*N*,*N*-dimethylsulfamate) was obtained via the modified procedure previously described in the literature [38]. For a detailed scheme, see Figure 6.

To a stirred solution of sodium hydride (60% dispersion in oil, 23.4 mg, 0.59 mmol) in anhydrous *N*,*N*-dimethylformamide (DMF; 2 mL) at room temperature, 5-hydroxyuracil (75 mg, 0.58 mmol) was added. The mixture was left in an ultrasonic bath under an inert gas atmosphere at 36 °C for 5 min. After the hydrogen emission stopped, *N*,*N*-dimethylsulfamoyl chloride (84 mg, 0.59 mmol) was added. Subsequently, the reaction mixture was vigorously stirred for 24 h, at 36 °C and poured into water (10 mL) and the final product was purified with semi-preparative HPLC (Shimadzu, LC 20AD, Kyoto, Japan,) with a UV detector (SPD M20A) which was set at 265 nm. For the separation of analytes, a C18 column (Phenomenex, Gemini, Warszawa, Poland; 150 × 10 mm, 5 µL in particle size and 100 Å in pore size) and linear-gradient flow (4 mL min^−1^) of 0–50% phase B in 30 min (mobile phase A: 0.1% formic acid, 1% acetonitrile and B: 80% acetonitrile) at 25 °C were used. The product was obtained as a white solid (2 mg) in 1.5% yield. Sodium hydride, 5-hydroxyuracil, *N*,*N*-dimethylsulfamoyl chloride, and anhydrous DMF were purchased from Sigma-Aldrich (Poznan, Poland).

^1^H NMR (400 MHz, DMSO, Appendix A) δ 11.49 (brs, 1H, NH), 11.14 (brs, 1H, NH), 7.72 (s, 1H, CH), 2.90 (s, 6H, CH_3_). HRMS (Appendix A), m/z: [M + H]^+^ calcd for C_6_H_9_N_3_O_5_S 236.0336; found 236.0656. UV spectrum (water; Appendix A), λ_max_: 268 nm. The ^1^H NMR spectrum was recorded on a Bruker AVANCE III HD, 400 MHz spectrometer (Billerica, MA, USA). Chemical shifts are reported in parts per million relative to the residual signal of DMSO-d_6_ (2.51 ppm). The MS measurements were done with the use of a TripleTOF 5600^+^ (SCIEX, Framingham, MA, USA), and the UV spectrum was recorded on a Dionex UltiMate 3000 System (Waltham, MA, USA), with a diode array detector.

### 3.3. Computational Methods

In order to obtain the lowest energy geometry of methylated SU, the hydrogens of the amino group were replaced with the methyl groups in the most stable conformer of SU [22] and the obtained geometry was optimized at the M06-2X [39]/aug-cc-pVTZ [40] level of theory. No geometrical constraints were used for the geometry optimization. All the geometries were geometrically stable, which was confirmed by the analysis of harmonic frequencies (all force constants were positive). This geometry and the geometries of all remaining reactants were optimized (and subjected to the frequency calculations) and used, in turn, for the calculations of thermodynamic thresholds related to the anions observed in the crossed electron-molecular beam experiments. The thresholds (ΔG) were calculated as the difference between the Gibbs free energies of products and substrate in their electronic ground states [21,22]. The latter were obtained using the rigid rotor-harmonic oscillator approximation for T and p equal to 298.15 K, 1 atm and 386.15 K and 5.33 × 10^−11^ atm, respectively. The latter pair of temperature and pressure corresponds to the actual conditions of the CEMB experiment. The pressure correction to the G value for the experimental pressure was applied using Equation (11) [41]:G_5_._33_ × _10_^−11^_atm,T_ = G_1atm,T_ + TS_trans;1atm,T_ − TS_5_._33_ × _10_^−11^_atm,T‘_(11)
where G_p,T_ is the free enthalpy at the pressure p and temperature T, and S_trans;p,T_ denotes translational entropy at the pressure p and temperature T.

The AEA_E_ (adiabatic electron affinity (AEA) in the electronic energy (E) scale) was calculated as energy difference between the optimized neutral and its corresponding anion in their optimum geometries (see Equation (12)).
AEA_E_ = E_neut_ − E_anion_(12)

The TNI anions formed as primary anionic states in the crossed electron molecular beam experiments are metastable states (resonance anions). Therefore, they cannot be described by Self-Consistent Field methodology, which is based on a variational principle. In order to estimate the position of a chosen resonance, we employed a variant of stabilization methods [42]. Namely, we increased by a fractional amount, Δq, the nuclear charge of the atomic center belonging to the bond that is cleaved in the experiment in order to artificially stabilize the anionic state [43,44]. Then, we calculated the electron binding energy, D, as a difference between the energy of the neutral and that of anion and extrapolated it to Δq → 0. It is worth noting that Δq was relatively large since the ab initio method leads to reliable results only for the electronically bound states and for the extrapolation procedure we considered only these points where the anion was electronically stable. Thus, the DMSU neutral geometry was optimized in the gas phase at the MP2/aug-cc-pVTZ level of theory. Then, for each considered resonance, the antibonding orbital, responsible for detaching the appropriate fragment of the molecule, was searched. The desired SOMO (singly occupied molecular orbital) orbitals were obtained via increasing the nuclear charge (Δq) at the appropriate carbon atom (to enable the antibonding orbital localized at a C-N bond) or oxygen atom (to find the antibonding orbital localized at an O-C or O-S bond) in the DMSU anion radical. For a range of nuclear charges, the difference between the neutral (E_neu_) and anion radical (E_anrad_) electronic energies (E_neu_ − E_anrad_ = D) was calculated at the MP2/cc-pVDZ level of theory. Extrapolation of binding energy, D, to Δq equal to zero let us estimate the resonance energy (E_R_), i.e., the position of a resonance.

All the calculations were carried out with the Gaussian09 [45,46] and Gaussian16 [47] suite of programs.

## 4. Conclusions

Dissociative electron attachment studies have been performed in the gas phase using a crossed electron molecular beam experimental setup to investigate the fragmentation pathways of DMSU due to low-energy electron attachment between 0–12 eV. Our findings showed similar fragmentation channels as in our previous study to SU. The mechanism of bond cleavages within this class of modified uracil derivatives seems to be mostly preserved regardless of the site. We observed (UO-H)^−^ as the most abundant anion following S-O bond cleavage. Overall, we observed 10 fragment anions with fragmentation pathways involving single and multiple bond cleavages. In agreement with calculations, most of the DEA channels are energetically opened at the electron energy as low as 0 eV. The role of methylation in DEA to DMSU was previously investigated for the nitroimidazole molecule [48,49,50]. For 1-methyl-4-nitroimidazole and 1-methyl-5-nitroimidazole, corresponding to a methylation of nitroimidazole at the N1-H site, DEA results showed the quenching of low energy resonances especially below 2 eV [48,49]. In this present study of DMSU, such drastic behavior was not observed. However, the ratio of anion yield from the uracil-5-yl moiety and from the substituent is increased for DMSU. On the other hand, DEA to SU led to the formation of fragment anions that were linked to dissociation pathways leading to the formation of OCN^−^, C_2_H_2_N_2_O_2_^−^ and C_3_H_3_N_2_O_2_^−^ within the degraded uracil-5-yl moiety. Those anions were below the limit of detection in the present study. Another alternation by methylation was found for the fragment anion SO_2_^−^, which exhibited here a sharp peak close to 0 eV peak instead a feature at ~1 eV like in SU [22]. In addition, this anion also shows a suppression of the peak intensity between 4–8 eV previously observed in SU. Quantum chemical calculations were used to determine the thresholds and adiabatic electron affinities of the neutrals. Using a charge stabilization method for the calculations of resonances, we predict the formation of the NH^−^ rather than CH_3_^−^ which are isobaric anions with mass 15 u.

## Figures and Tables

**Figure 1 ijms-22-02344-f001:**
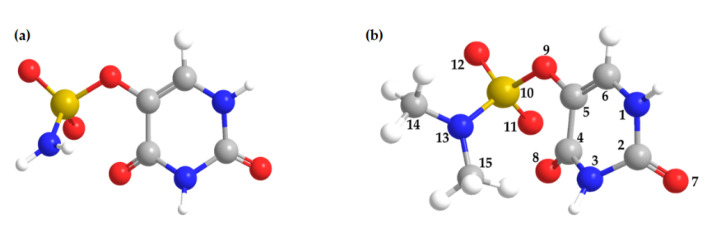
Molecular structure of (**a**) uracil-5-yl *O*-sulfamate (SU) and (**b**) uracil-5-yl *O-*(*N*,*N*-dimethylsulfamate) (DMSU).

**Figure 2 ijms-22-02344-f002:**
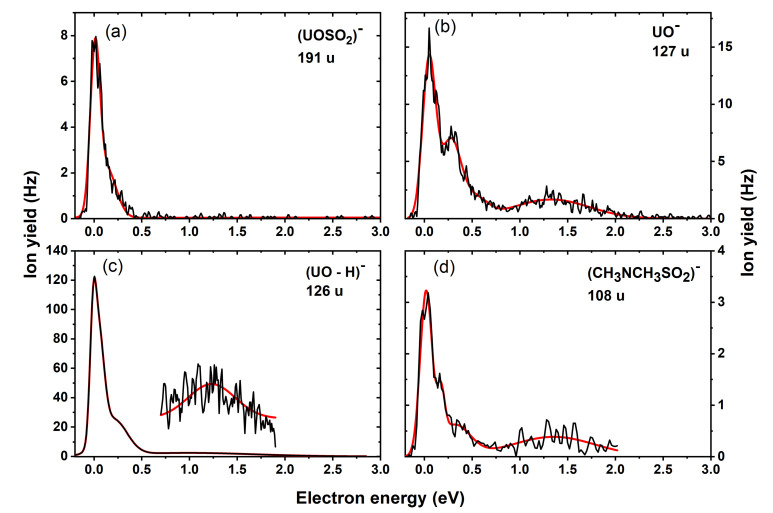
Anion efficiency curve for the fragment anions (**a**) UOSO_2_^−^, (**b**) (UO)^−^, (**c**) (UO-H)^−^ and (**d**) ((CH_3_)_2_NSO_2_)^−^ formed upon dissociative electron attachment (DEA) to DMSU.

**Figure 3 ijms-22-02344-f003:**
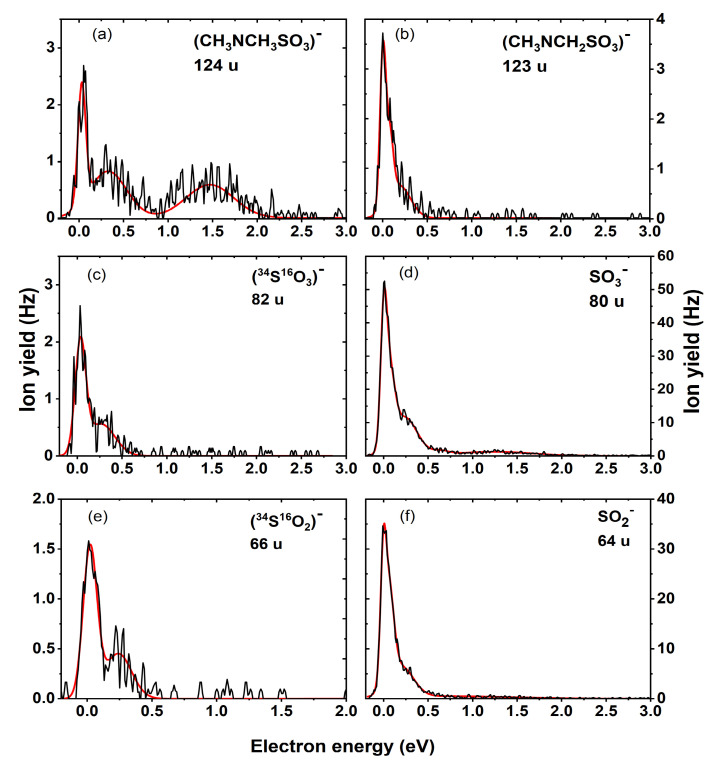
Anion efficiency curve for the fragment anions (**a**) CH_3_NCH_3_SO_3_^−^, (**b**) CH_3_NCH_2_NSO_3_^−^ (**d**) SO_3_^−^, and (**f**) SO_2_^−^, and for the isotopes (**c**) ^34^S^16^O_3_^−^ and (**e**) ^34^S^16^O_2_^−^ formed upon DEA to DMSU.

**Figure 4 ijms-22-02344-f004:**
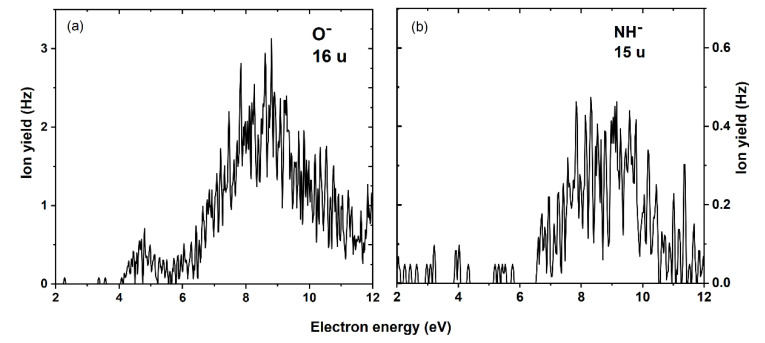
Anion efficiency curve for fragment anions (**a**) O^−^ and (**b**) NH^−^ formed upon DEA to DMSU.

**Figure 5 ijms-22-02344-f005:**
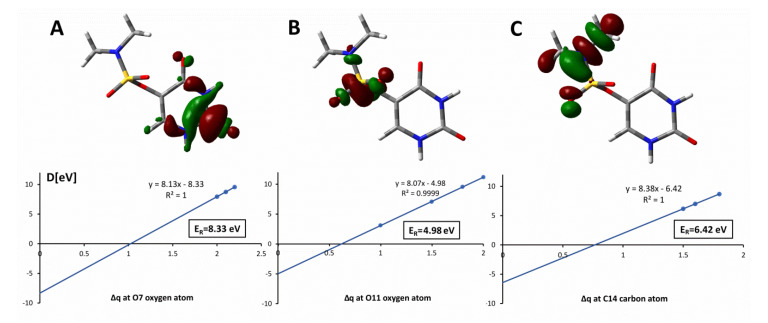
Binding energy, D, plotted at the stabilizing charge, Δq, values for which the anion is electronically stable and extrapolated to Δq → 0. Singly occupied molecular orbital (SOMO) visualized with the contour value of: 0.05 (a_0_)^−3/2^ for Δq = 2.1 (A panel); 0.08 (a_0_)^−3/2^ for Δq = 1.5 (**B**) and 0.05 (a_0_)^−3/2^ for Δq = 1.6 (**C**). E_R_ stands for the position of resonance related to the release of: O7 (**A**), O11 (**B**) or the C(14)H_3_ methyl group (**C**).

**Figure 6 ijms-22-02344-f006:**
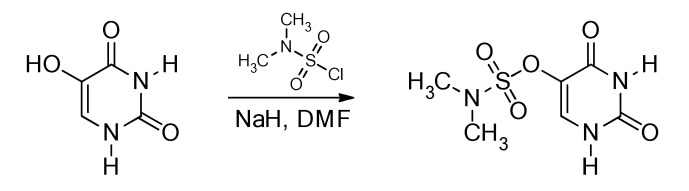
Synthesis of uracil-5-yl *O*-(*N*,*N*-dimethylsulfamate).

**Table 1 ijms-22-02344-t001:** Summary of fragment anions in terms of mass, structural assignments and their corresponding maxima of peaks, as well as the thermodynamic thresholds and adiabatic electron affinities determined upon electron attachment to DMSU.

**Mass** (**u**)	**Anion**	**Maxima of Peak Positions** (**eV**)				**Thresholds** (**eV**) **5.33 × 10^−11^ atm**			**AEA_E_** (**eV**)
		**1.**	**2.**	**3.**	**4.**	**Exp.**	**Calc.**		**Calc.**
						**386.15 K**	**298.15 K**	**386.15 K**	
191	UOSO_2_^−^	0	0.1			~0	−1.41	−1.78	4.11
127	UO^−^	0	0.3	1.3		~0	−0.60	−0.95	2.40
126	(UO-H)^−^	0	0.2	1.2		~0	−1.07 ^a^	−1.41 ^a^	4.84^a^
124	(CH_3_)_2_NSO_3_^−^	0	0.3	1.4		~0	−0.30	−0.64	4.66
123	((CH_3_)_2_NSO_3_-H)^−^	0	0.1			~0	−1.45	−1.79	4.88
108	(CH_3_)_2_NSO_2_^−^	0	0.2	0.3	1.4	~0	−0.97	−1.32	2.75
82	H_2_SO_3_^−^	- *				- *	−0.48	−1.18	−0.45 (0.01 ^b^)
80	SO_3_^−^	0	0.3	1.3		~0	−1.75	−2.09	5.09
66	H_2_SO_2_^−^	- *				- *	0.14	−0.55	−0.59 (0.08 ^b^)
64	SO_2_^−^	0	0.3			~0	−0.31	−0.66	1.43
16	O^−^	4.7	8.8			~4	2.61 ^c^	2.32 ^c^	1.41
15	CH_3_^−^	8.5				~6	3.46	3.12	0.001

* Experimentally not detected but investigated by computational methods. ^a^ For N1-H site; alternative site N3-H thresholds: −0.43/−0.78 eV at 298.15/386.15 K. ^b^ the G2MP2 [27] method of chemical accuracy was used for calculation of AEA_E_ for H_2_SO_3_ and H_2_SO_2_. ^c^ For O12 site; alternative sites O7: 4.86/4.57 eV, O8: 5.12/4.83 eV, O11: 2.76/2.45 eV at 298.15K/386.15K. ^d^ Corresponds to NH^−^ (see text).

## Data Availability

The data presented in this work are available on request from the corresponding authors.

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
