# Peer review of "Electron-Induced Decomposition of Uracil-5-yl O-(N,N-dimethylsulfamate): Role of Methylation in Molecular Stability"

_ijms, 2021, doi:10.3390/ijms22052344_

Round 1

Reviewer 1 Report

The authors have studied dissociative electron attachment (DEA) processes in the interaction of low-energy electrons with DMSU (C6H9N3O5S) molecules in gas phase. Appling crossed electron-molecular beam experiments resulted in the experimental detection of ten fragment anions with most of them formed within the energy range below 2 eV. The experimental results are supported by quantum chemical calculations on the thermodynamic thresholds of fragmentation reactions and resonance positions.

A gas phase study on the interactions of the proposed molecules with low-energy electrons enables the necessary information about the fragmentation pathways and the mechanism of anion formation in the DEA process to be obtained. Such information is relevant for our understanding of the mechanism of radiosensitization triggered by low-energy electrons.

The authors briefly described used methods in the study: a hemispherical electron monochromator coupled with a quadrupole mass analyzer under high vacuum conditions; the synthesis of the target molecules; and computational methods.

As the results fragment anions in terms of mass, structural assignments and their corresponding maxima of peaks, as well as the thermodynamic thresholds and adiabatic electron affinities determined upon electron attachment are given. For almost all molecular fragmentations upon DEA to DMSU, the major ion yield was observed near 0 eV. In two cases the maximum efficiencies were found at higher energies: for mass 16 u at 4.7 eV and 8.8 eV (interpreted as a release of the oxygen anion, O−) and for mass 15 u at 8.5 eV (interpreted as a release of CH3− or NH− anion). By using charge stabilization calculations and predicting the resonance positions corresponding to the release of CH3-, at much lower energy of about 6 eV, the authors concluded that the dominant fragmentation is in NH− anion.

Author Response

We thank the reviewer for reading our manuscript and the favorable report. Since the reviewer does not request any revision, no modifications of the manuscript were done in view of his/her report.

Round 2

Reviewer 2 Report

no further action is required